# Questioning approaches to consent in time critical obstetric trials: findings from a mixed-methods study

Elizabeth Deja ![ORCID],[1] Andrew Weeks,[2,3] Charlotte Van Netten,[4] Carrol Gamble,[5] Shireen Meher,[6] Gillian Gyte,[7] Tina Lavender ![ORCID],[8] Kerry Woolfall[1]

[1]Department of Public Health, Policy and Systems, University of Liverpool, Liverpool, UK
[2]Department of Women's and Children's Health, University of Liverpool, Liverpool, UK
[3]Liverpool Women's NHS Foundation Trust, Liverpool, UK
[4]Liverpool Clinical Trials Centre, University of Liverpool, Liverpool, UK
[5]Health Data Science, University of Liverpool, Liverpool, UK
[6]Birmingham Women's Hospital, Birmingham, UK
[7]National Childbirth Trust, London, UK
[8]Department of Clinical Sciences, Liverpool School of Tropical Medicine, Liverpool, UK

**Correspondence to**
Dr Elizabeth Deja;
e.deja@liverpool.ac.uk

## ABSTRACT

**Objective** Trial legislation enables research to be conducted without prior consent (RWPC) in emergency situations, yet this approach has rarely been used in time-critical obstetric trials. This study explored views and experiences of antenatal recruitment and consent and RWPC in an emergency intrapartum randomised clinical trial.

**Design** Embedded, mixed-methods study within a trial, involving questionnaires, recorded recruitment discussions, interviews and focus groups in the first 13 months of trial recruitment (December 2020–January 2022).

**Setting** COPE is a double-blind randomised controlled trial, comparing the effectiveness of carboprost or oxytocin as first-line treatment of postpartum haemorrhage.

**Participants** Two hundred and eighty-six people (190 women/96 birth partners), linked to 198/380 (52%) COPE recruits participated in the embedded study. Of these, 272 completed a questionnaire (178 women/94 birth partners), 22 were interviewed (19 women/3 birth partners) and 16 consent discussions with 12 women were recorded. Twenty-seven staff took part in three focus groups and nine staff were interviewed.

**Results** Participants recommended that information about the study should be more accessible antenatally for those who wish to be informed. Most women and staff did not think it would be appropriate to seek consent during pregnancy or early labour as it may cause 'unnecessary panic' and lead to research waste, as most women would not become eligible. There was support for the use of RWPC as COPE interventions are used in standard clinical practice and viewed as low risk. Women who were approached about the trial while having a postpartum haemorrhage also supported RWPC as they could not recall research discussions.

**Conclusions** Findings support the use of RWPC for time-critical interventions, and raise questions about the appropriateness of other commonly used consent pathways, including antenatal consent and verbal assent.

For the original protocol for the Carboprost or Oxytocin Postpartum haemorrhage Effectiveness trial (COPE), see online supplemental material 1.

## STRENGTHS AND LIMITATIONS OF THIS STUDY

⇒ A comprehensive mixed-methods design, including questionnaires and in-depth interviews from key stakeholders including trial participants, birth partners and staff; this allowed for triangulation of data and depth of insight.

⇒ Insight was gained from women and birth partners who experienced informed consent through antenatal recruitment, as well as verbal assent during postpartum haemorrhage and research to be conducted without prior consent through emergency recruitment.

⇒ Data were collected until there was thematic saturation and to the point of information power, which is the point when data are deemed to address the study aims and sample specificity, such as experience relevant to the study aims and sample diversity.

⇒ The consent rate for the COPE trial was high (n=363/380, 96%), which limited opportunities to gain insight into the views and experiences of women who declined to take part.

⇒ Interviews were limited to participants who could speak English.

## INTRODUCTION

The Royal College of Obstetricians and Gynaecologists (RCOG) provides guidance on obtaining consent in time-critical perinatal research and recommends that the approach should vary according to the individual underlying risk of being eligible for participation.[1] However, there is no evidence to support this strategy, or the quoted levels of risk, which means the design of clinical trial consent processes may not be in line with the expectations or priorities of women and stakeholders.

Recruiting women to intrapartum research studies is complex.[2 3] Several strategies have been used to balance the need to gain truly informed consent with the desire not to unnecessarily distress or overmedicalise a natural process.[4] Predominantly, informed consent is sought from women before trial enrolment, however, this may not always

be possible in intrapartum research, for example, if a woman lacks capacity (e.g., is unconscious). Legislation makes provisions for an alternative decision maker to be involved in enrolment decisions. This is usually a relative (e.g., legal representative) or doctor who is independent of the research. However, involving an alternative decision-maker requires time for information exchange and discussion and thereby this is not possible where there is an urgent need to treat within a medical emergency. In such urgent care situations, women or their representative may have very little time or capacity to consider study information and make an informed decision about participating in research. Research has shown that staff may have ethical and legal concerns about enrolling women into a trial without written informed consent.[5] Some studies include an antenatal consent process to help offset such concerns. For uncommon conditions or events, this means approaching potentially large numbers of women who may never become eligible. This can be costly and may overburden women with information, or cause unnecessary worry for women, who may never experience the trial condition.[6 7]

Houghton *et al* conducted a qualitative study exploring women's views and experiences of consent in an obstetric emergency in the World Maternal Antifibrinolytic trial (WOMAN), which explored treatments for postpartum haemorrhage (PPH).[8] The WOMAN trial included options for informed consent and to randomise women without their prior consent if a relative or doctor was not present to consent on their behalf. Regardless of the consent approach, and whether or not they consented or declined, women struggled to recall recruitment conversations and were often unable to differentiate between clinical and research discussions. No single consent pathway was preferred. Findings may have been limited by poor recall due to a stressful birth experience and interviews being conducted more than a year after trial recruitment took place.

Trials legislation enables research to be conducted without prior consent (RWPC, also known as deferred consent) when there is an urgent need to treat a patient for the purposes of the trial.[9] This involves administering trial interventions to eligible patients without delay and then, once the emergency situation has passed, researchers seek consent for ongoing involvement in the study.[10 11] Although research has shown support for RWPC in paediatric and neonatal settings[12–15] there has been limited use of this approach in time critical obstetric trials. Sweeney et al's [7] pilot study explored women's views and experiences of RWPC within a postpartum cluster trial and showed support for the approach in a small sample of women.

COPE (The Carboprost or Oxytocin Postpartum haemorrhage Effectiveness Study) is a double-blind, randomised controlled trial (RCT) comparing the effectiveness of carboprost and oxytocin as first-line treatments of PPH. As part of an embedded study, our research aimed to explore women, birth partner and clinician views on the

various approaches to recruitment and consent to inform the ongoing RCT and future time critical obstetric trials.

## METHODS
### Setting
The COPE trial was designed with two consent processes: researchers could seek prospective antenatal consent for women at increased risk of PPH, or women could be recruited without prior consent during a PPH (see figure 1). The latter method is unusual in a labour setting and required assessment of feasibility and patient acceptability.

### Study design
We conducted an embedded mixed-methods study in all sites open during the first year of COPE trial recruitment (December 2020–January 2022). We used previous research[10 14 16–18] to develop participant information sheets (online supplemental material 2), questionnaires (online supplemental material 3) and topic guides (online supplemental material 4).

We collected questionnaires and conducted interviews with women and birth partners approached for participation in COPE. The questionnaire followed the same format as those used in similar studies[9 11 13] consisting of demographic questions, Likert scale questions from the Decision Making Control Instrument[14] and free text responses, taking on average five minutes to complete. Interview topic guides included questions on birth experiences, COPE information, approaches to recruitment, willingness to participate and views on trial acceptability

For clinical and research staff, we undertook focus groups and interviews and audio recorded trial consent discussions with patients and birth partners (if applicable). The staff focus group and interview topic guide explored experiences of recruitment and consent during the first year of recruitment, acceptability of the COPE trial, site training, screening, administering the COPE intervention, documentation, medication and logistics of running the trial.

### Patient and public involvement
Design: Before finalising the research protocol, the consent process went through extensive consultation with parents and public both in person and online with consumer groups.

Conduct: A trial consumer panel was formed, led by an experienced consumer representative and trial management group member (GG), who provided critical input and represented the participants' views throughout the study

Dissemination: GG is an author on this paper. The paper will be disseminated to participants through the COPE online media. In addition, an infographic of the findings will be sent directly to participants who requested a copy of the findings on their consent forms.

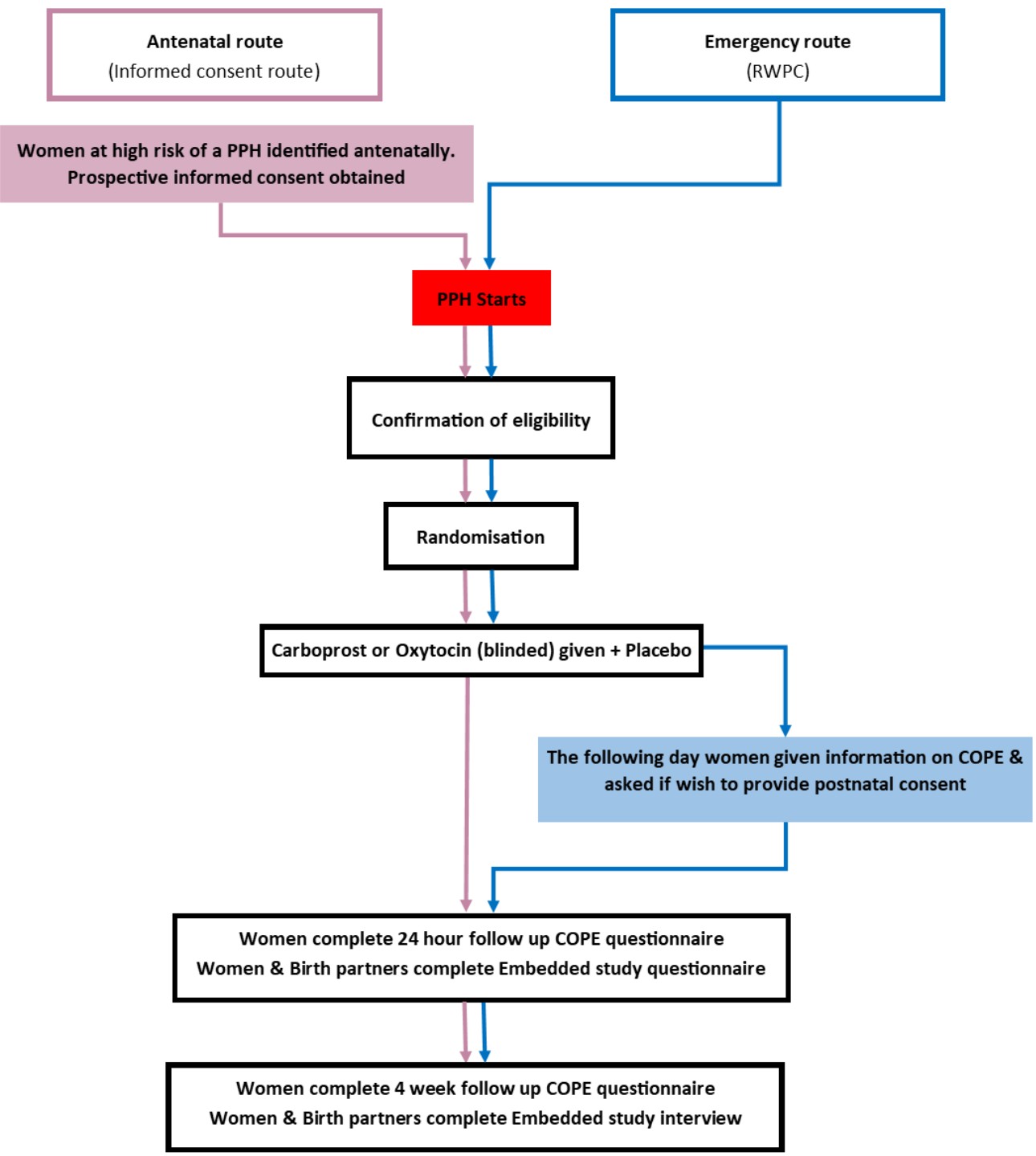

**Figure 1** The COPE trial RCT consent process. Key: PPH, postpartum haemorrhage; RCT, randomised controlled trial; RWPC, research to be conducted without prior consent.

## Women and birth partner recruitment and conduct

All women and their birth partners approached to take part in the COPE trial during its first year were eligible to take part in the embedded study if they spoke English. This included those who declined trial participation. When the COPE trial was broached by site staff, women and birth partners were informed of the embedded study and written consent was sought to complete a questionnaire and/or take part in a telephone/online interview with the COPE researcher (ED, PhD, female, psychologist, research associate) at a convenient time within approximately a month. Two pilot sites also sought consent to audio record consent discussions, which took place on a hospital ward within 24 hours of randomisation.

Eleven English sites recruited to COPE during the first year. Staff at all sites provided women and birth partners with questionnaires to complete immediately following the consent discussion with preaddressed envelopes to be returned to ED. ED contacted those who consented to interview in sequential order (by receipt of a consent form), stratifying by site and role (women/birth partner). Priority was given to those who had declined consent for the COPE trial and/or agreed to audio recordings. Based on previous studies,[10 14 19] we anticipated that 20–25 interviews would be needed to reach thematic saturation, where additional data do not lead to any new major themes identified during analysis. Researchers were also looking for high levels of 'information redundancy'[20–22] and information power, the point when data are deemed to address the study aims and sample specificity, such as experience relevant to the study aims and sample diversity.[20 23]

Due to COVID-19 restrictions, all interviews took place over the telephone. Respondent validation was used to add unanticipated topics to the topic guide as interviewing and analysis progressed.[16] A £30 shopping voucher was given to participants after the interview to thank them for their time.

## Staff recruitment and conduct

Initial findings from women and birth partner interviews and questionnaires were used to develop staff focus groups and interview topic guides (see online supplemental file 4). Coinvestigators at the first four open sites were contacted regarding the capacity to hold an online focus group using Microsoft Teams or Zoom. The research nurse and coinvestigators disseminated invitations to all staff involved in the conduct of the trial, including participant information sheets and consent forms to be completed before the focus group. Staff members that could not make the focus groups and/or were at other sites were invited to take part in an online interview.

## Analysis

Digital audiorecordings were transcribed verbatim by a professional transcription company (UK Transcription, Brighton) and anonymised. Qualitative analysis of interviews, focus groups, audiorecorded recruitment discussion data and open response questionnaire data were interpretive and iterative[24] using a reflexive thematic analysis approach[24 25] (see table 1). NVivo V.12 software (QSR International, Melbourne, Australia) was used to assist in the organisation and coding of data. Data from the parent and staff questionnaires were cleaned and entered into SPSS V.24.0 (IBM). Descriptive statistics are presented with percentages. ED and KW (PhD, female, social scientist, professor) analysed and synthesised the data, drawing on the constant comparative approach.[26 27]

## RESULTS
## Participants

During the first year of COPE trial recruitment, 382 women were screened for trial participation. A total

| Table 1 | Approach to qualitative data analysis |
| --- | --- |
| **Phase** | **Description** |
| 1. Familiarising with data | ED read and re-read transcripts noting down initial ideas on themes. |
| 2. Generating initial codes | Data-driven themes and concepts were identified by ED through line-by-line coding, discussions with KW and notes from phase 1. ED also developed the data-coding framework using a priori codes identified from the project aims, topic guides and past relevant experience of conducting trials methodology research |
| 3. Developing the coding framework | KW coded 20% of the interview transcripts using and refining the initial coding frame. |
| 4. Defining and naming themes | Following review by ED and KW coding frames were subsequently developed and ordered into themes (codes) within the NVivo database. (Investigator triangulation) |
| 5. Completion of coding of transcripts | ED completed coding interview transcripts in preparation for write-up. |
| 6. Quantitative data analysis | Descriptive statistics were conducted on questionnaire and survey data. |
| 7. Data synthesis | ED and KW synthesised qualitative and qualitative data drawing on a constant comparative approach. This involved identifying themes in each data set and reflecting on the study aims ensuring key findings and recommendations were relevant to the COPE trial design (catalytic validity). |
| 8. Stakeholder feedback | ED presented initial findings and sought feedback from a variety of key stakeholders; Including 'how to COPE' research nurse working groups, The study advisory group and management groups both of which have public, patient representation (respondent validation). |
| 9. Producing the report | Final discussion and development of selected themes occurred during the write-up phase. |

COPE, The Carboprost or Oxytocin Postpartum haemorrhage Effectiveness Study.

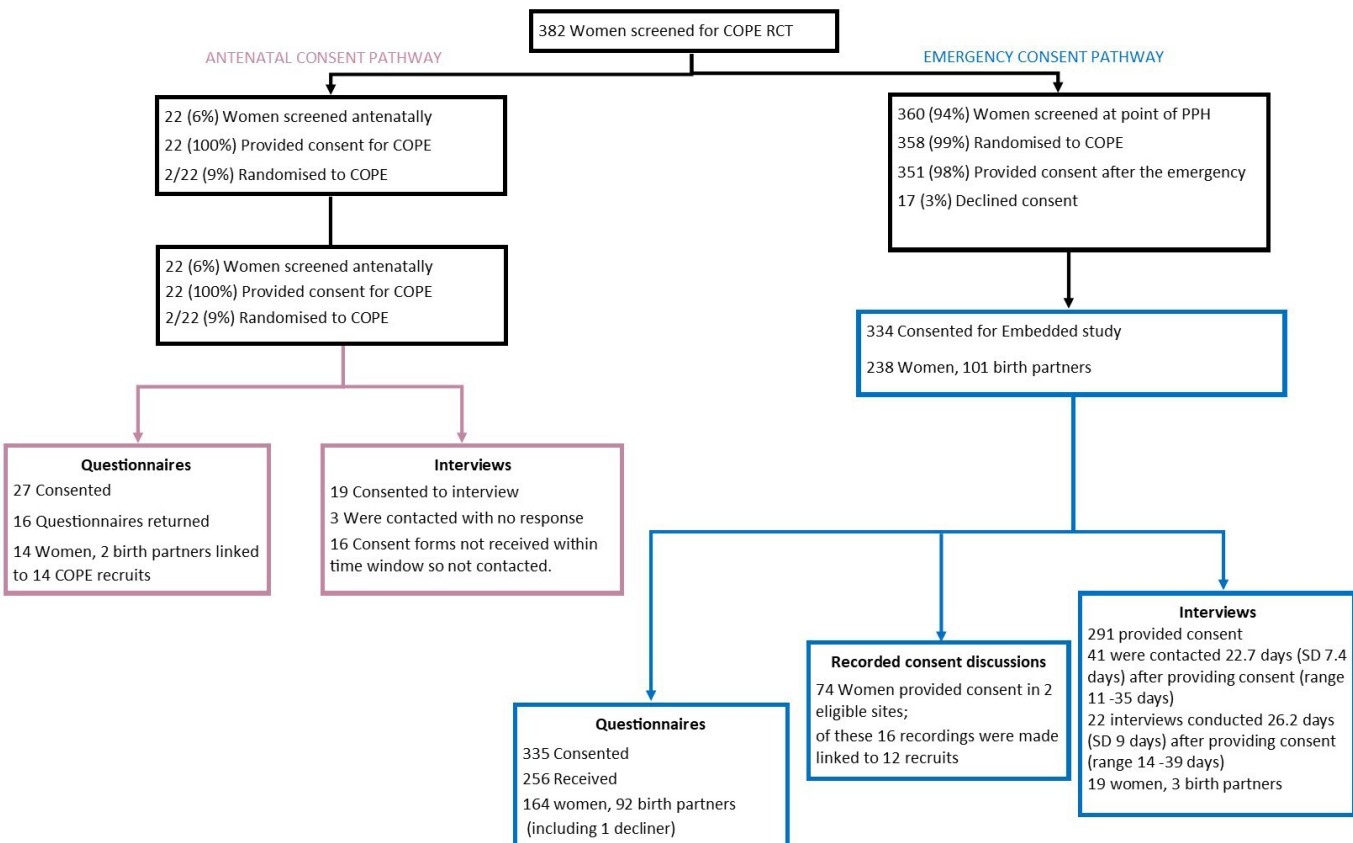

**Figure 2** Embedded study recruitment. PPH, postpartum haemorrhage; RCT, randomised controlled trial.

of 366 individuals (n=260 women/106 birth partners) gave consent to at least one part of the embedded study. Of these, 191 women and 96 birth partners provided data, linked to 199/380 (52%) COPE recruits at 9/11 (75%) sites (see figure 2). Within these, 16 consent discussions with 12 women (mean 9 min per woman, range 4.1–19.6) on the emergency pathway were recorded, 272 questionnaires (178 women/94 birth partners) and 22 telephone interviews (19 women/3 birth partners) (mean of 31.6 min, SD 9.3, range 17–49) were competed.

The women who completed the questionnaires had a mean age of 30 years old (range 18.4–42.9 years) and 41/178 (23%) had experienced a PPH in a previous pregnancy. The birth partners that competed the questionnaire had a mean age of 31.9 years (range 21.9–64.9 years) and 17/94 (18%) had themselves or had a partner who had experienced a PPH in a previous pregnancy. The 11 sites opened at different times across the 13 months of data collection. Four sites were open during the first 6 months, with the others opening between months 7 and 12, two in the last month. To prevent over-representation, the initial two sites were closed to the embedded study after 7 months as thematic saturation had been reached for both questionnaires and interviews. The other sites remained open to ensure information power was achieved.

The first four open sites took part in the staff component of the embedded study. A total of 36 staff self-identified as research midwives (n=10), midwives (n=4), anaesthetists (n=2), obstetricians (n=14), principal investigators (n=2), ward managers (n=1) and research leads (n=3). Of these, 27 took part in 3 focus groups at 2 sites (mean 28 min, range 31–52) and 9 in interviews (mean 50 min, range 38–103).

Of the women and birth partners who completed a questionnaire, most 162/272 (61%) stated they first heard about COPE after treatment for PPH, while 42/272 (16%) indicated they heard about the trial when bleeding prior to treatment and 24/272 (9%) heard during an antenatal appointment. The remaining 15% 'could not remember' (29/272, 11%), 'other' (4/272, 2%) or did not respond to this question (11/272, 4%).

## Antenatal recruitment: the importance of awareness

Women, birth partners and practitioners had mixed views on the use of antenatal recruitment to the COPE trial. All of the women and birth partners who were recruited antenatally (n=16) and/or indicated in the questionnaire that they 'first heard about COPE during an antenatal appointment' (questionnaire item) (n=8) stated they were 'satisfied with the consent process' (questionnaire item, 24/24, 100%).

The majority of women who were recruited in the emergency pathway, as well as staff involved in recruitment, believed that antenatal COPE recruitment may: 'be a bit overwhelming and cause mams to panic unnecessarily'

(P05, Women, Interview) for women who do not have a PPH and those that do:

> No, definitely after. Because you don't need that extra worry on top. And isn't it only 5% of women that haemorrhage anyway? It's such a low percentage. Even if you're telling the high-risk women, they're already concerned. Yes, I would say afterwards.
>
> P12, Women, Interview

> We didn't want to worry the women about PPH before the event because it might not happen to them. I think there was that fear of causing alarm about what might happen before it actually happens. You want her to look forward to the birth of her baby and not be worried about it. So, I think we were a little bit concerned that it might cause a bit of upset in the women unnecessarily
>
> P01, Staff, Interview

Staff highlighted that alongside the potential emotional burden, antenatal recruitment was practically and conceptually difficult. A couple of the staff described how the definition of high risk for PPH varied between hospitals, with conflicting definitions considered to be confusing and a barrier for antenatal recruitment:

> Some of our ladies that were classed as high-risk for a PPH weren't on your list as being high-risk for antenatal approaching. So, that caused a bit of confusion sometimes
>
> P01, Staff, Interview

Practical issues to antenatal recruitment also included the COVID-19 pandemic restrictions limiting the: '*number of face-to-face consultations… it decreases our chances of meeting people*' (P01, Focus Group 1), as well as being '*really time-consuming*' (P05, Staff, Interview) and wasteful as recruitment may not lead to randomisation as: '*actually they did not have postpartum haemorrhage*' (P02, Staff, Focus group 1). They were also concerned with the possibility, that for those who did have a PPH, there may not be a COPE trained member of staff available to administer the drugs. These challenges were reflected in trial recruitment data (see figure 2) with 22 antenatal consents resulting in just 2 randomisations.

Some staff were concerned about the possibility of accidentally recruiting a woman through the RWPC route who had previously declined consent antenatally. Despite processes to mitigate this risk being in place within the protocol, this possibility made some staff reluctant to recruit antenatally:

> Because you could accidentally randomise someone who maybe they will be a decline, but that person maybe hasn't quite realised that, several months down the line, or weeks or whatever, when they then come to the Labour Ward.
>
> P06, Staff, interview

Both women and staff reflected that the main benefit of antenatal recruitment: '*is that it increases awareness for the women*' (P04, Staff, Interview). Women recruited through the emergency route felt that raising awareness of the COPE trial antenatally would be sufficient rather than consent being sought at that point in time. Both women and staff felt that this could be achieved through parent-facing advertising: '*whether it's just posters and things like that*' (P04, Staff, Focus group 2) or an '*information leaflet on Badger* [electronic notes]' (P09, Staff, Interview). As the information would provide: '*the opportunity to know about it.[…]. they could read up more into for themselves.*' (P17, Women, Interview) for those that wish to do so.

### Assent discussion during PPH: situational incapacity

Some staff discussed broaching the COPE trial with women and/or birth partners during PPH when they felt it was appropriate, despite verbal assent not being specified within the COPE protocol. A total of 42/272 (15%) women and birth partners indicated in the questionnaire that 'a doctor or midwife first spoke to them about COPE while they were bleeding before treatment was given'. When describing their experience of labour and PPH, women's experiences varied greatly based on levels of blood loss, trauma, medical interventions, physical health, sedation and consciousness. All of these impacted their level of situational capacity and in turn their view on seeking verbal assent in COPE:

> I think I lost two and a half pints of blood. I just remember being very cold, very uncomfortable and frankly a little bit scared. Then I pretty much woke up four hours later and she [the baby] was already here.
>
> P07, Women, Interview

> I was nervous beforehand, but the staff were great. They just put my mind at rest and, yes, it just felt like it went really quickly. Everything seemed to go well. Yes, it was a really calm and positive experience.
>
> P15, Women, Interview

During interviews, the few women who recalled the COPE trial being briefly discussed during their PPH were largely indifferent to the research discussion. Due to the consequences of going through labour and having just given birth, they often stated they had not taken in any study information due to lack of capacity:

> Yes, I was fine with it. [being told about COPE] it was more beneficial that they came round after rather than explaining it fully in the [Caesarean] section room because in the theatre I wasn't thinking straight, so I couldn't tell you, I couldn't relay anything that they said in there.
>
> P13, Women, interview

In contrast, one woman who stated she wanted to be 'in control of her body' clearly recalled asking questions about the research and felt reassured to have all the

information about what was going on, including information about COPE:

> For me personally, I ask questions because I like to feel a little bit in control of my body, I like to know exactly what's happening. So, me asking questions and them being able to explain it there and then, that helped me come to terms with it, that helped calm me down really because I like to know step by step what is exactly happening.
>
> P05, Women, interview

Women who did not hear about the COPE trial during their PPH were asked their views on seeking verbal assent at that point in time. Most felt that discussing research would be inappropriate, particularly if there was a lot of blood lost or in the context of a traumatic birth. As the quotations below illustrate, one woman felt that she was dying during her PPH. Most stated they would have been emotionally and physically unable to understand or retain research information and make an informed decision:

> There was one point in theatre when I thought, "I actually am going to die." But aside from that they were amazing. Everyone in the theatre was cheering me on, being like, "Push. I dare you. Push." It was a bit like being at a football match.
>
> P20, Women, Interview

> In my opinion, labour is one of the most unplannable events that you can have. You can be completely low risk and end up requiring litres of blood. I think nobody is going to be retain that information at the time if they are haemorrhaging or having somebody place metal forceps in their vagina.
>
> P18, Women, Interview

Not all women were aware that they had a PPH and some felt that broaching the trial to seek verbal assent could negatively affect women's birth experiences and cause panic by drawing attention to the bleeding:

> I was completely unaware that I was having a bleed, whereas I think if they had made me aware in theatre that I was actually having a bleed and asked for my consent, I think I would have started panicking. I think that would have completely changed that really nice, calm memory I have got of giving birth. I think it is a good thing that they didn't ask for my consent right there and then, and then put me onto it.
>
> P15, Women, Interview

### Seeking consent when the emergency situation has passed

The majority (351/373, 94%) of all COPE participants during the first year were recruited through the RWPC route. In our sample, 96% of women and birth partners indicated they were satisfied (n=144/161) or indifferent (n=11/161) with 'the consent process for COPE' (Questionnaire Item). They described how it: '*made sense when I*

*read the information why it was done that way.*' (P21, Women, Interview) as they would not have been able to take in the information at the time and on the whole would prefer not to have it mentioned to them to before, or during labour. This support for RWPC appeared to be underpinned by their: '*trust in the people that are looking after you*' (P19, Women, Interview), that staff: '*wouldn't do it if it's going to put you at risk or anything, so it's fine.*' (P01, Women Interview). RWPC in COPE was also viewed as acceptable because both treatment arms were: '*approved in practice in its own right*' (P18, Women, Interview).

While all those interviewed were happy with the use of RWPC, interview and audiorecorded recruitment discussion data indicated that at the time of consent, commonly within 24 hours after birth, they were: '*shattered*' (P10, women, recruitment discussion recording), '*I'm really tired. I've just come over all tired*' (P12, Women, Recruitment discussion recording); '*My whole body feels really weak and lethargic.*' (P05, Women, Recruitment discussion recording) and: '*in quite a lot of pain still*' (P16, Women, Interview). Alongside their physical condition: '*there was just too much happening*' (P10, Women, Interview) with people: '*just popping in to make sure you're okay*' (P02, Staff, Recruitment discussion recording), women and baby checks, clinical care and looking after the baby. Consequently, they commented on how they had asked fewer questions than they might have done in another context. This was reflected in the audio recordings, which were short in length (mean 9 min, range 4.1–19.6) and predominantly consisted of the staff talking.

Despite the personal discomfort experienced at the point that the COPE trial was broached, during interviews women accurately recalled the purpose of the COPE trial and stated they had made an informed decision about taking part, with multiple opportunities to ask staff questions if they had wished. Recruitment discussion data indicated those who did ask questions were focused on clarifying details of their PPH: '*How much blood loss compared to…?*' (P05, Women, Recruitment discussion recording) and whether or not they had only received a placebo. During interviews, women reflected that their interactions with staff were clear and well handled:

> She came back and we still hadn't had a chance to read it. So, she left and then she came back later that day. She was great. It was all very clear. And she was very conscientious and understanding of the situation, just to give us space and time.
>
> P12, Women, Interview

After discussing all the possible types of consent with women and birth partners during interviews, many concluded that if there had been an option between antenatal and RWPC recruitment for COPE, their preference was for the RWPC/emergency route where: '*there was no consent given and they just gave it to me*' (P15, Women, Interview) due to the reasons outlined above.

Of the 6/161 (4%) who stated they were 'not satisfied' with the use of RWPC in the questionnaire, 4 were birth partners and two were mothers. They used the open response section of the questionnaire to explain that they would have preferred to give consent in advance because: '*Seeking consent after the fact is disconcerting*' (P60, Women, Matched, Questionnaire) and: '*consent takes away some of the good will we have towards the programme*' (P101, Birth Partner, Questionnaire). One birth partner felt that: '*it didn't seem an emergency and more of a 'just in case' treatment* (P60, Birth partner, Matched, questionnaire) therefore not meeting the criteria for RWPC. One mother was not happy to have received an unnecessary intervention (the placebo): '*Tell people what is going in their bodies, don't get me wrong it helped but do not like the dummy needle*' (P199, Women, Questionnaire).

Staff described how they initially had concerns about the use of RWPC as their: '*knee-jerk response was the ethics of how can you give drugs to somebody without their consent?*' (P01, Staff, Interview). It was also considered: '*a really daunting approach for us because we really didn't know how women would take it*' (P04, Staff, Interview). However, experience with this approach appeared to change opinions about the acceptability of RWPC in this setting: '*I think the more we got through the trial, I think, the better we all have felt about it.*' (P03, Staff, Focus group 3). Staff found RWPC acceptable in COPE because: '*the fact that the women are okay with it, puts to bed any worries that I had about how it would be perceived.*' (P09, Staff, Interview). As the trial progressed, staff also concluded that giving the PPH treatment without prior consent was closer to standard practice as they do not ask for consent when administering PPH drugs in emergency situations:

> If you have got a situation as a PPH, you don't go through all the different drugs. You don't say to a woman, 'We are giving this drug and this does this and that does that.' You don't go into great detail. You just say you are giving some medication and de-brief them afterwards because, in that emergency situation, you haven't got time (P08, Staff, Interview)

As also highlighted by women, staff described how RWPC was acceptable in the COPE trial because it involved approved medications and actively treating the PPH: '*it is not a placebo medication, they are getting an active medication as well*' (P01, Staff, Focus group 1). Therefore: '*we not exposing the patient to unnecessary harm because of the trial*' (P01, Staff, Focus group 3) and: '*In essence, they are going to get their usual care*' (P11, Staff, Focus group 1).

The main issue staff experienced with RWPC was timing. They needed to obtain consent for the 24-hour main COPE trial questionnaire as close to that time point as possible. This was only if women were not busy and had the capacity to do so.

Trial screening logs indicated that the 17 women who declined consent were due to 'not receiving treatment for PPH' (n=2); women 'not wishing to take part in research' (n=7), no translation services being available (n=2), too

much paper work' (n=1) and 'being too tired' (n=1). Or women providing verbal agreement, however, leaving hospital before written consent being obtained and then not returning the consent forms (n=4).

## DISCUSSION

Establishing the most appropriate consent pathway for intrapartum research with women who may be vulnerable, have fluctuating capacity and need emergency treatment for PPH is ethically and practically challenging.[2–4 6–8] This mixed-methods study provides insight into the experiences of women, birth partners and clinicians and their views on approaches to recruitment and consent in the COPE trial, including the use of RWPC, which is novel in obstetric trials.

Our findings suggest that seeking informed consent antenatally was considered acceptable to those women and birth partners who experienced this pathway in COPE. Despite the COPE trial being designed in accordance with RCOG guidance,[1] staff involved in recruitment, and women randomised to the trial through the emergency consent pathway questioned the ethics of consenting women who may never become eligible for the trial. They were concerned about causing unnecessary anxiety and information burden for women by asking them to consider something going wrong with their pregnancy, particularly for those that do not go on to have a PPH. This finding was also reported by Alvarez et al,[28] who raised concerns about how a desire to not cause distress may lead to selection bias, with staff choosing who to approach. Women and birth partners in our study prioritised information provision, such as leaflets or posters, about the trial in antenatal settings over an informed consent process. Providing such information to women at increased risk of PPH would provide the opportunity for consideration by those who wish to do so[6] and therefore, in line with the bioethical principle of autonomy, protecting their right to make their own decision about research participation.[29 30] Staff were also concerned about the potential for research waste, a concern validated when the first 22 research consent discussions in COPE led to just two randomisations. After the first year of the COPE trial, taking into consideration the findings of the embedded study and recruitment rates, the antenatal informed consent pathway was removed from the COPE protocol.

Despite verbal assent not being specified as part of the recruitment process in the COPE protocol, we found some clinicians did seek verbal assent for participation in the trial when women were having a PPH. The majority of women stated they were not in a position to have, or understand research discussions at that point in time. As reported by Sweeney et al and Houghton et al,[7 8] recall of such recruitment discussions was poor. We found that due to situational incapacity, verbal assent at the time of PPH was only viewed as acceptable in a minority of cases when women are seeking out information and appeared

to have capacity. However, assessing such capacity is extremely difficult[2][3] and, as our data suggest, clinicians did not always correctly judge if a woman had the capacity to discuss research. Women in our study expressed a clear preference to not be approached about research at this time.

To our knowledge, this is the first study to explore experiences of RWPC in an obstetric RCT. RWPC was viewed by women and birth partners as being logical, acceptable and appropriate as COPE interventions were considered low risk and time critical. This is in line with views of RWPC in paediatric emergency situations[10][13][19] and Sweeney et al's Pilot trial.[7] Staff had some initial apprehension about the use of RWPC. However, they were reassured by women's positive responses. Roper et al's[31] seven-step framework to enhance practitioner explanations and parental understandings of RWPC in paediatric emergency and critical care trials highlights the importance of explaining the condition before the consent discussion. This is particularly relevant to obstetric trials where women may have fluctuating, or no capacity, and therefore, not be aware of what has happened during birth. Indeed, there were both women and birth partners in our study who were unaware that a PPH had occurred when COPE was broached by staff. In line with guidance on RWPC,[11] we recommend that research staff should check the appropriate timing of the research discussion with the bedside staff and ensure the patient is aware of their clinical condition and history during childbirth before explaining how they were entered into a clinical trial. Research staff should also explain the reasons why consent could not have been sought prospectively.

### Strengths and limitations of this study

The study was strengthened by its mixed-methods design including multiple data collection methods at various time points from all key stakeholders including birth partners. Data triangulation is considered to increase confidence in the findings from the research.[32][33] The study included all sites open within the first year which provided sample variance (75% of sites) and data from all sites that opened in the first 11 months of COPE retirement. Giving insight into views on consent processes from a large proportion of COPE participants and birth partners linked to half (52%) of the COPE recruits.

This study has several limitations. First, we had restricted insight into the views and experiences of women who declined to take part in the COPE trial. However, 100% of women approached antenatally provided consent and 97.5% of women recruited via the emergency pathway provided consent, so there was limited opportunity to access the minority who declined. Screening logs indicated that reasons for declining were not related to the approach to recruitment and more about not wanting to take part in any research. An exception was one birth partner, whose questionnaire indicated they had declined due to not being satisfied with the RWPC process. Second, audiorecorded recruitment conversations were restricted to two sites. Although limited in quantity, the insight gained from the data we did obtain suggested they were often very brief conversations led by clinicians and corroborated the accounts given by both women and staff. These data further highlight the importance of assessing appropriate timing of RWPC discussion with women who have just had a traumatic childbirth. Third, there were no interviews with antenatal consent pathway recruits because of delays in obtaining and/or processing consent due to staffing shortages during the initial phase of the COVID-19 pandemic. Questionnaire data linked to 14 COPE recruits 14/22 (63%) who experienced this pathway provided insight into their satisfaction with the recruitment process they experienced, but this did not provide the depth of understanding that would be gained through an interview.

### CONCLUSION

Recruiting women to intrapartum research studies is practically and ethically challenging. Our findings support the use of RWPC for time-critical, low-risk interventions and it raises questions about the appropriateness of other commonly used consent pathways, including seeking informed consent antenatally and verbal assent during an obstetric emergency.

**Acknowledgements** Special thanks to all the families who took part in the research and to our patient partners for their contribution to the design, conduct and study findings.

**Contributors** KW led the study, which was embedded within the COPE Trial led by AW. ED, CG, GG, LT and AW were involved in the design of the study; or the acquisition, analysis or interpretation of data. ED collected the data. ED analysed the data with oversight from KW. ED and KW drafted the article and AW, LT and CG critically revised it for important intellectual content. All authors gave approval for the version to be published. KW and ED agree to act as guarantors and to be accountable for all aspects of the work in ensuring that questions related to the accuracy or integrity of any part of the work are appropriately investigated and resolved.

**Funding** The COPE trial, including this embedded study, was funded by the National Institute for Health Research (NIHR) Health Technology Assessment (HTA) (project number 16/16/06) and coordinated by the University of Liverpool.

**Disclaimer** The views and opinions expressed herein are those of the authors and do not necessarily reflect those of the HTA Programme, NIHR, NHS or the Department of Health. The funders had no role in the collection, analysis and interpretation of data or in the decision to submit this article for publication.

**Competing interests** NIHR, HTA programme grant payments were made to the institutions of all authors to support the conduct of this study. ED and CVN employment roles were funded by the NIHR HTA programme grant payments made to their institutions.

**Patient and public involvement** Patients and/or the public were involved in the design, or conduct, or reporting, or dissemination plans of this research. Refer to the Methods section for further details.

**Patient consent for publication** Not applicable.

**Ethics approval** This study involves human participants and was approved by West Midlands - Coventry & Warwickshire Research Ethics Committee (18/WM/0227). Participants gave informed consent to participate in the study before taking part.

**Provenance and peer review** Not commissioned; externally peer reviewed.

**Data availability statement** No data are available. No data are available. Although data relevant to the study are included in the article, no raw data are available. The datasets generated and analysed for this study are not publicly available due to participants being from a small, specialised population, which creates ethical and privacy concerns about being too identifiable.

**ORCID iDs**
Elizabeth Deja http://orcid.org/0000-0002-3626-4927
Tina Lavender http://orcid.org/0000-0003-1473-4956

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
