## [Reviewer comments · BMJ Open]

ARTICLE DETAILS

TITLE (PROVISIONAL)	Questioning approaches to consent in time critical obstetric trials: findings from a mixed methods study
AUTHORS	Deja, Elizabeth; Weeks, Andrew; Van Netten, Charlotte; Gamble, Carrol; Meher, Shireen; Gyte, Gillian; Lavender, Tina; Woolfall, Kerry

VERSION 1 – REVIEW

REVIEWER	Elden, Helen Goteborgs Universitet, Health and Care Sciences
REVIEW RETURNED	23-Nov-2023

GENERAL COMMENTS	I wish to express my appreciation for the authors' compelling and well-crafted manuscript. However, I have several inquiries and suggestions. Firstly, there is a lack of information on the baseline characteristics of the informants. Currently, only the profession and number of informants are presented. No details are provided about the women or birth partners, except for their numerical representation. Secondly, despite the study being presented as a mixed methods study involving both questionnaires and interviews, only the results from the interviews are presented. I recommend including the results from the questionnaires as well. Additionally, I am eager to obtain more details about the qualitative analysis, particularly the stages in the analysis process. Furthermore, I am interested in learning how the researchers addressed their pre-understanding. Lastly, I observe the absence of a discussion on trustworthiness, covering elements such as credibility, dependability, confirmability, and the transferability of the qualitative results.
---

REVIEWER	Walker, Shawn Imperial College London, Women's Health
REVIEW RETURNED	25-Nov-2023

GENERAL COMMENTS	This is a very valuable and excellently written article. The important findings are presented in a pragmatic way that will enable them to inform guidance and practice for time-critical obstetric trials. The methods are described very clearly. The design is appropriate and very rigorous. The findings are delivered in a balanced way, giving adequate voice to the dissenting opinions. A key strength of this study is its inclusion of data from women who experienced different recruitment strategies. This enabled some comparison of experiences and consideration of the benefits
---

	and drawbacks of each. The researchers have adequately accounted for the fact that there may be a variety of preferred approaches, rather than one that fits all. For example, some women may want access to antenatal information, despite a majority not feeling this is necessary. This is important because maternity services and research must seek to meet the needs of all, not just the majority. It is a drawback that only women who could speak English were included in interviews, although I understand why that would be the case. I recommend publication and have no substantial recommendations for revisions. I look forward to discussing this publication with maternity researchers in a journal club someday.
--	--

VERSION 1 – AUTHOR RESPONSE

Reviewer: 1 Dr. Helen Elden, Goteborgs Universitet	Dear Dr Elden thank you for your useful and detailed feedback. We hope that the changes we have made in response to your comments help clarify our methodology/ results.
I wish to express my appreciation for the authors' compelling and well-crafted manuscript. However, I have several inquiries and suggestions. Firstly, there is a lack of information on the baseline characteristics of the informants. Currently, only the profession and number of informants are presented. No details are provided about the women or birth partners, except for their numerical representation	We have added demographic information collected in the birth women/ birth partner questionnaire to the results including Information on Age and past experience of PPH. Professional roles and the sites they worked in were the key characteristics collected for this group. We are unable to name the sites or give additional information to ensure anonymity.
Secondly, despite the study being presented as a mixed methods study involving both questionnaires and interviews, only the results from the interviews are presented. I recommend including the results from the questionnaires as well.	Questionnaire data is included throughout the results section and woven into the qualitative data. for example "A total of 42/272 (15.4%) women and birth partners indicated in the questionnaire." or "In our sample, 96.3% of women and birth partners indicated they were satisfied (n= 144/161) or indifferent (n=11/161) with 'the consent process for COPE' (Questionnaire Item)." We have added additional questionnaire data to the beginning of the results section to show when participants first heard about the trial.

Additionally, I am eager to obtain more details about the qualitative analysis, particularly the stages in the analysis process.	Table 1 has been added that outlines the qualitative data analysis process.
Furthermore, I am interested in learning how the researchers addressed their pre-understanding.	In the questionnaire and interviews we asked participants 'when did a doctor or midwife first speak to you about cope?' (questionnaire) or when did you first hear about Cope? (interview) to assess their pre-understanding and also awareness of the trial via each recruitment pathway. We have added additional data from the questionnaire related to this 'pre-understanding' to the results
I observe the absence of a discussion on trustworthiness, covering elements such as credibility, dependability, confirmability, and the transferability of the qualitative results.	In the new Table 1 outlining the analysis process we have detailed how we addressed catalytic validity, respondent validation and Investigator triangulation. The use of data triangulation to also add trustworthiness to the data has been added to the strengths and limitations section of the discussion.
Reviewer: 2 Dr. Shawn Walker, Imperial College London, Imperial College Healthcare NHS Trust Comments to the Author:	
This is a very valuable and excellently written article. The important findings are presented in a pragmatic way that will enable them to inform guidance and practice for time-critical obstetric trials. The methods are described very clearly. The design is appropriate and very rigorous. The findings are delivered in a balanced way, giving adequate voice to the dissenting opinions. A key strength of this study is its inclusion of data from women who experienced different recruitment strategies. This enabled some comparison of experiences and consideration of the benefits and drawbacks of each. The researchers have adequately accounted for the fact that there may be a variety of preferred approaches, rather than one that fits all. For example, some women may want access to antenatal information, despite a majority not feeling this is necessary. This is	Dear Dr Walker Thank you for taking the time to review our manuscript and for your thoughtful comments. We agree that only including women who speak English was not ideal and have ensured that there is funding in place for a translator in our future studies. We will aim to assess the quality of such interviews to inform more inclusive trials methodology research.

important because maternity services and research must seek to meet the needs of all, not just the majority.

It is a drawback that only women who could speak English were included in interviews, although I understand why that would be the case.

I recommend publication and have no substantial recommendations for revisions. I look forward to discussing this publication with maternity researchers in a journal club someday.

VERSION 2 – REVIEW

REVIEWER	Elden, Helen Goteborgs Universitet, Health and Care Sciences
REVIEW RETURNED	28-Dec-2023
GENERAL COMMENTS	I approve of the author's revisions and comments and endorse the manuscript for publication in The BMJ Open.